# cT1N0M0 Esophageal Squamous Cell Carcinoma Invades the Muscularis Mucosa or Submucosa: Comparison of the Results of Endoscopic Submucosal Dissection and Esophagectomy

**DOI:** 10.3390/cancers14020424

**Published:** 2022-01-15

**Authors:** Ching-Ya Wang, Bo-Huan Chen, Cheng-Han Lee, Puo-Hsien Le, Yung-Kuan Tsou, Cheng-Hui Lin

**Affiliations:** 1Department of medicine, College of Medicine, Chang Gung University, Taoyuan 333, Taiwan; chingya1103@cgmh.org.tw (C.-Y.W.); spring03258@cgmh.org.tw (B.-H.C.); b9102011@cloud.cgmh.org.tw (C.-H.L.); b9005031@cloud.cgmh.org.tw (P.-H.L.); linchehui@cloud.cgmh.org.tw (C.-H.L.); 2Department of Medical Education, Chang Gung Memorial Hospital, Taoyuan 333, Taiwan; 3Department of Gastroenterology and Hepatology, Chang Gung Memorial Hospital, Taoyuan 333, Taiwan

**Keywords:** adjuvant therapy, chemoradiotherapy, endoscopic submucosal dissection, esophagectomy, squamous cell carcinoma, superficial esophageal cancer

## Abstract

**Simple Summary:**

Esophagectomy has been the standard treatment for superficial esophageal squamous cell carcinoma (SESCC) that invades the muscularis mucosa or submucosa. This retrospective study aimed to assess the effects of endoscopic submucosal dissection (ESD) combined with selective adjuvant chemoradiotherapy on these SESCCs by comparing the results of esophagectomy. We found no significant differences in overall survival, disease-specific survival, and progression-free survival between the two groups. In the ESD group, the procedure time, overall complication rates, and length of hospital stay were significantly reduced, but the rate of metachronous tumor recurrence was significantly increased. Therefore, ESD combined with selective adjuvant chemoradiotherapy can be an alternative treatment to esophagectomy for SESCC invading the muscularis mucosa or submucosa.

**Abstract:**

Background: Endoscopic submucosal dissection (ESD) combined with selective adjuvant chemoradiotherapy may be a new treatment option for cT1N0M0 esophageal squamous cell carcinoma (ESCC) invading muscularis mucosa or submucosa (pT1a-M3/pT1b). We aim to report the effectiveness of this treatment by comparing the results of esophagectomy. Methods: This retrospective single-center study included 72 patients with pT1a-M3/pT1b ESCC who received ESD combined with selective adjuvant chemoradiotherapy (*n* = 40) and esophagectomy (*n* = 32). The main outcome comparison was overall survival (OS). The secondary outcomes were treatment-related events, including operation time, complication rate, and length of hospital stay. Disease-specific survival (DSS) and progression-free survival (PFS) were also evaluated. Results: There were no significant differences in the rates of OS, DSS, and PFS between the two groups (median follow-up time: 49.2 months vs. 50.9 months); these were also the same in the subgroup analysis of pT1b ESCC patients. In the ESD group, the procedure time, overall complication rates, and length of hospital stay were significantly reduced. However, the metachronous recurrence rate was significantly higher. In a multivariate analysis, tumor depth and R0 resection were the independent factors associated with OS. Conclusions: ESD combined with selective adjuvant chemoradiotherapy can be an alternative treatment to esophagectomy for cT1N0M0 ESCC invading muscularis mucosa or submucosa.

## 1. Introduction

Esophageal squamous cell carcinoma (ESCC) has a poor prognosis, but early diagnosis and treatment can significantly improve survival [1]. Esophagectomy with lymph node (LN) dissection is traditionally the standard treatment for superficial ESCC (SESCC). However, despite improvements in surgical techniques and facilities, the morbidity and mortality associated with the surgery are still major concerns [2,3]. In addition, its impact on the long-term, health-related quality of life and physical health of surviving patients after esophagectomy is negative [4]. Therefore, if proved effective, a less invasive alternative to esophagectomy would be very valuable in the treatment of SESCC. In this case, endoscopic submucosal dissection (ESD) has recently become an acceptable alternative therapy for SESCC invasion of the epithelium (pT1a-M1) or lamina propia (pT1a-M2) [5,6]. However, the main limitation of ESD is that it can only remove the primary tumor, not the metastatic lymph nodes (LNs). Therefore, according to Japanese guidelines, SESCC that invades the muscularis mucosa (pT1a-M3) or submucosa (pT1b) is a relative/investigational indication due to some risks of LN metastasis [6].

Because the current diagnostic tools are not accurate enough to establish a good correlation between the cT-stage and the pT-stage, it is difficult to accurately assess the tumor depth before ESD [7,8]. Therefore, it is reasonable to offer diagnostic ESD for cT1N0M0 ESCC patients as an initial diagnostic and therapeutic modality [8]. On the other hand, compared with esophagectomy, definitive chemoradiotherapy (CRT) not only has a considerable overall survival (OS) rate for cT1N0M0 ESCC, but also has the advantage of organ preservation [9,10]. However, definitive CRT has two main limitations, namely a high local failure rate (19–29%) and complications related to dose escalation [11,12,13]. In this case, ESD combined with CRT would be an ideal solution. On the one hand, ESD can be used to remove the tumor to reduce the local failure rate. On the other hand, it can reduce the subsequent CRT dose to minimize the risk of complications [14]. That is, based on the pathological findings after ESD, selective CRT can be provided for patients at risk of subclinical LN (e.g., pT1a-M3/pT1b ESCC) [12,13,15,16,17,18]. Recently Minashi et al. reported that ESD combined with selective adjuvant CRT for cT1bN0M0 ESCC could achieve oncologic results that may not be inferior to esophagectomy, but there is a lack of comparative studies [19]. Therefore, we conducted this study to report the outcomes between cT1N0M0 and pT1a-M3/pT1b ESCC patients who received ESD combined with selective adjuvant CRT and esophagectomy.

## 2. Patients and Methods

### 2.1. Patients

Between January 2008 and December 2016, a total of 171 consecutive cT1N0M0 ESCC patients were identified from the computer database of our institution’s cancer registry. The study flowchart is shown in Figure 1. The following patients were excluded: (1) patients who did not receive ESD or esophagectomy (*n* = 37); (2) patients who became lost at follow-up after diagnosis or treatment (*n* = 18); (3) pT1a-M1/M2 after ESD or esophagectomy (*n* = 40); and (4) pT2 or deeper after esophagectomy (*n* = 3). Therefore, a total of 73 pT1a-M3/pT1b patients were enrolled: 49 patients received primary ESD, and 24 patients received primary esophagectomy. Nine of the forty-nine patients underwent adjuvant esophagectomy after primary ESD; one of the nine patients was excluded from the study because he underwent adjuvant esophagectomy more than 10 weeks (33 weeks) after ESD. The remaining 8 patients and the 24 patients with primary esophagectomy were classified into the esophagectomy group (*n* = 32). The remaining 40 patients with primary ESD were classified into the ESD group (*n* = 40). The study was approved by our institutional review board (IRB number: 202001305B0).

### 2.2. Clinical Staging

The tools used for clinical staging before the primary treatment included endoscopic ultrasound (EUS), chest computed tomography (CT) scan, and integrated fluorodeoxyglucose positron emission tomography/CT (PET/CT). EUS was the main modality to determine the invasion depth of the ESCC. On EUS, cT1a was the ESCC involving only mucosa; cT1b was the ESCC involving submucosa; and cT1 was determined when neither cT1a nor cT1b could not be determined. Chest CT and PET/CT scans were carried out to identify possible LN or distant metastasis.

### 2.3. ESD and Esophagectomy Procedures

The detailed procedures for ESD were similar to those described in our previous reports [20,21]. Usually, only those patients with cT1aN0M0 ESCC were indicated for ESD in our institution. However, ESD was also considered for cT1(b)N0M0 ESCC patients who were not suitable or refused primary esophagectomy. Most primary ESD procedures (44/49 or 89.8%) were carried out between 2013 and 2016. For esophagectomy, the main method was thoracoscopic-laparoscopic esophagectomy combined with two-field (mediastinal and abdominal) LN dissection. The number of dissected LNs depended on the surgeon, and the median of LNs as determined by pathological examination was 25 (range, 2–49). About half (13/24 or 54.2%) of the primary esophagectomy cases were completed between 2013 and 2016. The management of the treatment-naïve patients was decided by a dedicated multidisciplinary meeting involving chest surgeons, gastrointestinal endoscopists, oncologists, radiation oncologists, radiologists, and pathologists.

### 2.4. Pathological Staging

All ESD specimens were pinned on a cork and fixed with 10% formalin. The specimens were cut into parallel 2 mm serial sections in ESD specimens and 5 mm in surgical specimens. All specimens were stained with hematoxylin and eosin. In pathological studies, all specimens were examined for the resection margins, depth of tumor invasion, lymphovascular invasion (LVI), and histological type. For ESD specimens with tumors invading the submucosa, they were divided into two layers according to the depth of invasion: SM1 was determined when the tumor infiltrated the submucosa up to 200 μm; SM2 was determined when the tumor invaded more than 200 μm [22]. Complete resection was defined as en bloc resection without cancer cells at all resection margins, and it was expressed as R0.

### 2.5. Esophagectomy after Primary ESD

After ESD, we recommended additional therapy (esophagectomy or CRT) for the following patients with: (1) positive vertical resection margins; (2) the presence of LVI in the resection specimens; and/or (3) any cancer with invasion ≥ SM2. However, whether they would receive adjuvant therapy and which adjuvant therapy was chosen depended on the patient’s comorbidities or the patient’s wishes.

Eight patients received esophagectomy ≤ 10 weeks (median 4.6 weeks) after the primary ESD due to: pT1a-M3 with positive vertical resection margin (*n* = 1), pT1b-SM1 with LVI (*n* = 1), pT1b-SM2 (*n* = 2), pT1b-SM2 with positive vertical resection margin (*n* = 2), pT1b-SM2 with LVI (*n* = 1), and pT1b-SM2 with positive vertical resection margin and LVI (*n* = 1). Two of the patients (25%) were LN-positive at the esophagectomy (each patient had one positive node).

### 2.6. Adjuvant Therapy in the ESD Group

In the ESD group, 19 patients (47.5%) met the above-mentioned criteria for adjuvant CRT (all patients were pT1b, including 3 patients with positive vertical resection margin and 2 patients with LVI). However, of the 19 patients, only 7 (36.8%) received adjuvant therapy: 6 received CRT, and 1 received radiotherapy (RT) alone. The other patients received close follow-ups. Systemic chemotherapy aimed at treating patients’ synchronous cancer (such as head and neck cancer) was not regarded as adjuvant therapy for ESCC in this study. For patients who underwent adjuvant CRT, combined chemotherapy with 5-fluorouracil and cisplatin and concurrent radiotherapy (total dose of 5000 cGy/25 fractions) were commonly used.

### 2.7. Follow-Up

The patients’ follow-up data were updated in March 2020. During the follow-up period, for patients in the ESD group, an esophagogastroduodenoscopy (EGD) with image-enhanced endoscopy (narrow band images and/or Lugol’s staining) was performed every 3–6 months, and a chest CT scan was performed every 6–12 months. Patients in the esophagectomy group had an EGD and CT scan every 6–12 months. Metachronous recurrence was defined as a neoplasm that was detected at the esophagus other than the resection site (i.e., the scar area or surgical anastomosis) at least 6 months after the ESD/esophagectomy. Local recurrence was defined as a neoplasm that was detected at the scar area/anastomosis of the esophagus. Regional/distant recurrence was defined when LN metastasis or a new malignant lesion outside the esophagus was detected. For ESD patients who did not receive adjuvant therapy, a relapse in regional LN within 12 months after ESD was regarded as pre-existing LN metastasis.

### 2.8. Outcome Comparisons

The primary outcome was OS, which was defined as the time from ESD/esophagectomy to death from any cause. Secondary outcomes were (1) variables related to the treatment, including procedure time, complications, and length of hospital stay; (2) recurrence or metastasis at the end of follow-up; and (3) disease-specific survival (DSS) and progression-free survival (PFS), which were defined as the time from ESD/esophagectomy to death from ESCC recurrence and the time from ESD/esophagectomy to the occurrence of local/LN/distant metastasis, respectively.

### 2.9. Statistical Analyses

Continuous variable data were represented by the median and range; categorical variables used a number (%). For comparisons, the Mann–Whitney U test was used for continuous variable data, and the chi-square test or Fisher’s exact test was used for categorical variables. The Kaplan–Meier estimator was used to estimate the survival probability between groups, and the log-rank test was used to compare survival outcomes. In order to analyze the independent factors related to OS in clinical and pathological variables, univariate and multivariate Cox proportional hazards models were used. Only those variables with *p* < 0.1 in univariate analyses were entered into a multiple regression analysis for the OS. A two-tailed *p*-value < 0.05 was considered statistically significant. Statistical analysis was performed using SPSS software (version 18.0; SPSS, Inc., Chicago, IL, USA).

## 3. Results

### 3.1. Patient Characteristics and Tumor Features

#### 3.1.1. Patient Characteristics

Demographic data and tumor features of the study population are listed in Table 1. In the demographic data, age, sex, and performance status were not significantly different between the two groups. Comorbidities (including diabetes mellitus, hypertension, end-stage renal disease) were not significantly different, except that more patients with liver cirrhosis were observed in the ESD group (17.5% vs. 0%, *p* = 0.015). Compared with the esophagectomy group, more patients in the ESD group had a history of past but cured cancer (50% vs. 25%, *p* = 0.031). Synchronous cancer had a prevalence of 17.5% in the ESD group and 31.2% in the esophagectomy group (*p* = 0.172). The clinical T-stage was significantly different: the prevalence of cT1a/cT1b/cT1 in the ESD group was 77.5%/12.5%/10% of the patients, and 43.8%/40.6%/15.6% of the patients in the esophagectomy group (*p* = 0.009). Excluding nine patients with an uncertain clinical stage (i.e., cT1 stage), the sensitivity, specificity, positive predictive value, and negative predictive value of EUS for detecting pT1b ESCC were 31.7%, 77.3%, 72.2%, and 37.8%, respectively.

#### 3.1.2. Tumor Features

Regarding tumor characteristics, the median tumor length in the ESD group was significantly longer (30 mm vs. 23.5 mm, *p* = 0.004). However, this result might not be clinically meaningful because the tumor length in the ESD group was determined by Lugol staining, while the esophagectomy group was determined by gross appearance. There was no significant difference in tumor location and tumor differentiation between the two groups. Post-treatment pathology showed that in the ESD group, the prevalence of pT1a-M3 cancer was significantly higher (47.5% vs. 12.5%), while the incidence of pT1b cancer was lower (52.5% vs. 87.5%, *p* = 0.001). The frequency of LVI in the specimens of the ESD group was significantly lower than that of the esophagectomy group (5% vs. 28.1%, *p* = 0.007). Four patients (12.5%) in the esophagectomy group had pathological LN metastasis (three of them experienced LVI).

### 3.2. Procedure-Related Outcomes

The primary procedure-related events are summarized in Table 2. In terms of therapeutic effects, the R0 resection rate in the ESD group was 87.5%, and the R0 resection rate in the esophagectomy group was 96.9% (*p* = 0.217). The ESD procedures were performed by three endoscopists. The R0 resection rates of endoscopist A and endoscopist B + C were 90.9% (30/33) and 56.2% (9/16) (*p* = 0.008), respectively. The median procedure time (122.5 min vs. 402 min, *p* < 0.001) and median hospital stay (9 days vs. 25 days, *p* < 0.001) in the ESD group were significantly reduced. The types of procedure-related complications were quite different between the two groups: esophageal stricture (defined as the presence of dysphagia requiring endoscopic dilatation) was the most common complication (6/40 or 15%) in the ESD group, while pulmonary complication requiring chest tube (or pig-tail catheter) drainage for more than 10 days was the most common complication (11/32 or 34.4%) in the esophagectomy group. Major complications of the ESD group and the esophagectomy group (defined as ≥ grade 3 in the Clavieen–Dindo classification) occurred in 15% and 31.3% or patients, respectively (*p* = 0.099). No adverse event requiring surgical intervention was observed in the ESD group. There was no 30-day mortality in either group.

### 3.3. Outcomes of the Patients

The outcomes of the patients are listed in Table 2. The median follow-up period was 49.2 months (range, 2–93.8 months) in the ESD group and 50.9 months (range, 7.7–119.1 months) in the esophagectomy group (*p* = 0.683). The incidence of metachronous recurrence was significantly higher in the ESD group (30% vs. 0%, *p* = 0.001), and most (9/12 or 75%) metachronous recurrences were high-grade dysplasia. All patients with a metachronous recurrence were cured by ESD. However, the LN and/or distant recurrence rate was lower in the ESD group, although it did not reach statistical significance (10% vs. 25%, *p* = 0.09). The Kaplan–Meier survival curves for OS, DSS, and PFS are presented in Figure 2a. At the end of the follow-up, no significant differences were found in the OS rate (*p* = 0.419), DSS rate (*p* = 0.436) and PFS rate (*p* = 0.176) between the ESD and esophagectomy groups. The 1-, 3-, and 5-year OS rates were 85.0%, 75.0%, and 66.7% versus 84.4%, 65.6%, and 53.3% for the ESD and esophagectomy groups, respectively. The 1-, 3-, and 5-year DSS rates were 96.6%, 96.6%, and 88.8% versus 95.8%, 91.5%, and 82.5% for the ESD and esophagectomy groups, respectively. The 1-, 3-, and 5-year RFS rates were 87.5%, 72.5%, and 67.0% versus 84.4%, 59.4%, and 52.7% for the ESD and esophagectomy groups, respectively. We performed a subgroup analysis of pT1b ESCC. There were 21 patients and 28 pT1b ESCC in the ESD and esophagectomy groups, respectively. The results are shown in Figure 2b. At the end of the follow-up, no significant differences were found in the OS rate (*p* = 0.712), DSS rate (*p* = 0.589), and PFS rate (*p* = 0.373) between the ESD and esophagectomy groups. In addition, we analyzed the results between patients who underwent ESD combined with RT/CRT (*n* = 7) and esophagectomy (*n* = 32). The results are shown in Figure 2c. At the end of the follow-up, no significant differences were found in the OS rate (*p* = 0.999), DSS rate (*p* = 0.920), and PFS rate (*p* = 0.735) between the ESD combined with RT/CRT and esophagectomy groups.

### 3.4. Factors Associated with Overall Survival

Table 3 lists the univariate and multivariate analysis results of OS in all study populations. In a univariate analysis, the factors associated with OS were synchronous cancer (Yes vs. No, HR = 2.91, 95% CI: 1.37–6.16, *p* = 0.005), tumor depth (pT1b vs. pT1a-M3, HR = 3.61, 95% CI: 1.25–10.41, *p* = 0.017), R0 resection (No vs. Yes, HR = 2.55, 95% CI: 0.96–6.75, *p* = 0.06 (marginally significant)), and LN/distant recurrence (Yes vs. No, HR = 3.45, 95% CI: 1.59–7.45, *p* = 0.002). Treatment method was not significantly associated with OS (esophagectomy vs. ESD, HR = 1.36, 95% CI: 0.64–2.91, *p* = 0.421). In the multivariate analysis, tumor depth (pT1b vs. pT1a-M3, HR = 3.09, 95% CI: 1.04–9.16, *p* = 0.042) and R0 resection (No vs. Yes, HR= 3.51, 95% CI: 1.27–9.65, *p* = 0.015) remained the factors associated with OS.

## 4. Discussion

ESD has been widely proved to be safe and effective for the treatment of SESCC with a low risk of LN metastasis [5,23]. Because prospective comparative studies would be neither feasible nor ethical, several retrospective studies compared the results of ESD and esophagectomy for the treatment of SESCC [8,24,25]. All studies concluded that ESD is not inferior to esophagectomy from the perspective of both surgery and oncology [8,24,25]. However, ESD for pT1a-M1/M2 SESCC has been proved to be sufficiently radical; only pT1a-M3/pT1b SESCC represents the gray area in treatment [6,26]. Therefore, in the present study, we only enrolled cT1N0M0 ESCC patients who were confirmed to have pT1a-M3/pT1b after the treatment. Our study revealed that the oncologic results are comparable between ESD combined with selective adjuvant therapy and esophagectomy.

In this study, the R0 resection rate was one of the independent factors associated with OS in the multivariate analysis. We found that the R0 resection rate was significantly different between the endoscopists. According to our experience, inexperienced ESD endoscopists tended to dissect the superficial submucosa to avoid perforation, leading to a higher positive deep resection margin. Therefore, to achieve better oncology results, endoscopic resection of SESCC (particularly cT1b cancer) is best reserved for qualified ESD endoscopists. Tumor depth was another independent factor associated with OS, which had been reported in several other studies [25,27]. Interesting, not only in this study but also in the literature, the treatment method (ESD vs. esophagectomy) was not a factor associated with OS [24,25,27]. Dubecz et al. reported that the 5-year OS rate of pT1 ESCC was 62%, and there was no significant difference between N0 and N + patients (61% and 65%, *p* = 0.591) [28]. In the present study, compared with the ESD group, the incidence of LN/distant recurrence in the esophagectomy group did not decrease. The possible reasons are: (1) There were significantly more pT1b ESCC patients in the esophagectomy group. Patients with cT1a ESCC tended to receive ESD rather than esophagectomy; (2) for the cT1b case, if the EUS endoscopist judged the case as a superficial submucosal infiltration at the multidisciplinary meeting, the patient would be recommended to receive ESD; (3) after ESD, patients at risk of lymph node metastasis or recurrence received adjuvant therapy (CRT or esophagectomy). Among the eight patients who underwent esophagectomy after primary ESD, two had pathological lymph node metastasis after the operation; and (4) more patients in the esophagectomy group had synchronous cancer. When the detectable LN/distant metastasis could not be confidently attributed to a primary cancer, we attributed the metastasis to ESCC. In this case, esophagectomy might not prevent late tumor recurrences in this study [29].

Diagnostic ESD, or ESD combined with selective adjuvant therapy, has several advantages in treating cT1N0M0 ESCC patients. First, most patients who follow this strategy can keep their esophagus intact. Organ-preserved therapy improves not only morbidity and mortality but also the quality of life [29]. The risk of subclinical LN for cT1bN0M0 ESCC patients is 20–27%, indicating that some patients do not require adjuvant therapy after ESD [10,30]. Dermine et al. reported that close follow-ups may be an alternative to esophagectomy after ESD of early esophageal cancer with a predicted high risk of subclinical LN [29]. Second, compared with esophagectomy, ESD is a safer treatment, as it is characterized by shorter operation time, lower incidence and severity of complications, and shorter hospital stays [25,27]. Third, through selective adjuvant therapy, patients with the risk of subclinical LN may not be undertreated by ESD [19,31]. Fourth, diagnostic ESD has no negative impact on surgical/oncologic outcomes once adjuvant esophagectomy is required [8]. However, a higher metachronous recurrence rate seems to be the disadvantage of this organ-preserved therapy [31]. Fortunately, as this study shows, most of this sequel can be treated endoscopically.

They were some limitations in this study. First, the retrospective design with a small number of cases might introduce a selection bias, especially in tumor depth, which is a prognostic factor for SESCC [5,22]. Therefore, we conducted a subgroup analysis of pT1b ESCC, and the comparison of prognostic results was similar to the analysis of the entire study population. Second, there has been no consensus on the selection criteria for patients receiving adjuvant therapy after ESD. In this study, one of the three criteria (≥SM2 invasion) for adjuvant therapy after ESD was not strictly enforced if these patients had achieved R0 resection plus no LVI. Further studies are needed to determine the selection criteria for adjuvant therapy. Third, which type of adjuvant therapy after ESD should be used was not standardized. In our recent review article, we found that ESCC ≥ SM2 invasion combined with LVI should be treated by adjuvant esophagectomy rather than CRT because of the high risk of tumor relapse if adjuvant CRT is given in this case [31]. A prospective study is needed to clarify this issue [32].

## 5. Conclusions

In conclusion, this single-center retrospective cohort showed that ESD combined with adjuvant CRT provides long-term OS, DSS, and PFS comparable with esophagectomy in patients with cT1N0M0 ESCC invading deep mucosa (pT1a-m3) or submucosa (pT1b). ESD has a lower incidence of surgery-related events compared with esophagectomy. Therefore, ESD combined with selective adjuvant CRT can be an alternative treatment to esophagectomy for these patients.

## Figures and Tables

**Figure 1 cancers-14-00424-f001:**
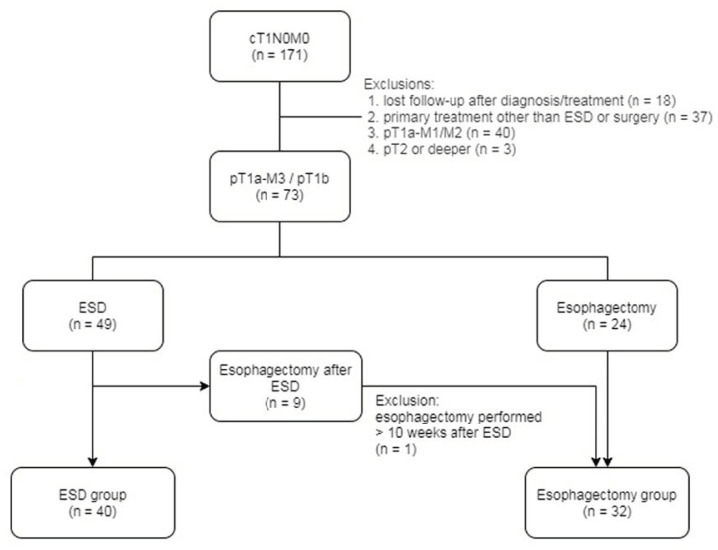
Flowchart of the study. Abbreviations: ESD: endoscopic submucosal dissection.

**Figure 2 cancers-14-00424-f002:**
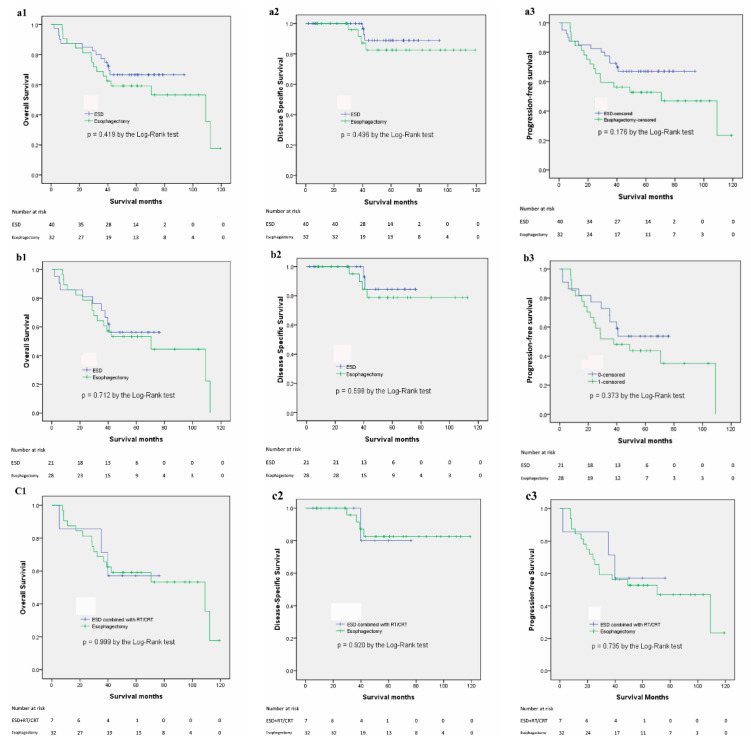
Kaplan–Meier survival curves for the endoscopic submucosal dissection and esophagectomy groups. (**a**): all study population; (**a1**): overall survival; (**a2**): disease-specific survival; (**a3**): progression-free survival. (**b**): a subgroup of patients with pT1b diseases; (**b1**): overall survival; (**b2**): disease-specific survival; (**b3**): progression-free survival. (**c**): endoscopic submucosal dissection combined with radiotherapy/chemoradiotherapy versus esophagectomy: (**c1**): overall survival; (**c2**): disease-specific survival; (**c3**): progression-free survival. Abbreviations: ESD: endoscopic submucosal dissection; RT: radiotherapy; CRT: chemoradiotherapy.

**Table 1 cancers-14-00424-t001:** Demographic data and tumor characteristics.

	ESD Group (*n* = 40)	Esophagectomy Group (*n* = 32)	*p*-Value
Median age, years (range)	55 (43–92)	54 (40–72)	0.946
Male, *n*	39 (97.5%)	32 (100%)	1
Comorbidities			
Liver cirrhosis, *n*	7 (17.5%)	0	0.015
Diabetes mellitus, *n*	5 (12.5%)	4 (12.5%)	1
Hypertension, *n*	12 (30%)	5 (15.6%)	0.154
ESRD, *n*	0	1 (3.1%)	0.444
Past cancer history but cured, *n*	20 (50%)	8 (25%)	0.031
Synchronous cancer, *n*	7 (17.5%)	10 (31.2%)	0.172
Performance status			1
ECOG 0/1, *n*	39 (97.5%)	32 (100%)	
ECOG ≥ 2, *n*	1 (2.5%)	0	
Clinical T-stage			0.009
cT1a, *n*	31 (77.5%)	14 (43.8%)	
cT1b, *n*	5 (12.5%)	13 (40.6%)	
cT1, *n*	4 (10%)	5 (15.6%)	
Tumor characteristics			
Median length, mm (range)	30 (11–82)	23.5 (9–50)	0.004
Location			0.533
Upper third, *n*	7 (17.5%)	7 (21.9%)	
Middle third, *n*	19 (47.5%)	11 (34.4%)	
Lower third, *n*	14 (35%)	14 (43.7%)	
Involvement ≥ 3/4 circumference, *n*	12 (30%)	NA	-
Invasion depth on post-treatment pathology			0.001
pT1a-M3, *n*	19 (47.5%)	4 (12.5%)	
pT1b, *n*	21 (52.5%)	28 (87.5%)	
LVI, *n*	2 (5%)	9 (28.1%)	0.007
Lymph node metastasis, *n*	NA	4 ^†^ (12.5%)	
Differentiation			0.327
Well and Moderate, *n*	23 (57.5%)	22 (68.8%)	
Poor, *n*	17 (42.5%)	10 (31.2%)	

Abbreviations: ESD: endoscopic submucosal dissection; ECOG: Eastern Cooperative Oncology Group; ESRD: end-stage renal disease; NA: not available; LVI: lymphovascular invasion. ^†^ Three patients experienced LVI.

**Table 2 cancers-14-00424-t002:** Procedure-related events and outcomes of the patients.

	ESD Group (*n* = 40)	Esophagectomy Group (*n* = 32)	*p*-Value
Procedure-related events			
Median procedure time, min (range)	122 (28–390)	408 (267–759)	<0.001
R0 resection, *n*	35 (87.5%)	31 (96.9%)	0.217
R0 resection plus no LVI, *n*	33 (82.5%)	23 (71.9%)	0.281
Major complications, *n*	6 (15%) ^‡^	10 (31.3%)	0.099
Grade IIIa ^†^, *n*	6 (15%) ^‡^	6 18.8%)	-
Grade IIIb or more ^†^, *n*	0	4 (12.5%)	-
Perioperative mortality, *n*	0	0	1
Median hospital stay, days (range)	9 (5–38)	25 (15–62)	<0.001
Adjuvant therapy, *n*	7 (17.5%)	0	0.015
Chemoradiotherapy, *n*	6 (15%)	0	-
Radiotherapy alone, *n*	1 (2.5%)	0	-
Outcomes of the patients			
Median follow-up period, month (range)	49.2 (2.0–93.8)	50.9 (7.7–119.1)	0.683
Metachronous lesions, *n*	12 (30%)	0	0.001
Squamous cell carcinoma, *n*	3	0	-
High-grade dysplasia, *n*	9	0	-
LN and/or distant recurrence	4 (10%)	8 (25%)	0.09
All-cause mortality, *n*	13 (32.5%)	16 (50%)	0.132
Disease-specific mortality, *n*	3 (7.5%)	4 (12.5%)	0.692

Abbreviations: ESD: endoscopic submucosal dissection; LN: lymph node; LVI: lymphovascular invasion; ^†^ The Clavien–Dindo classification; ^‡^ All were esophageal stricture needed endoscopic dilatation.

**Table 3 cancers-14-00424-t003:** Univariate and multivariate analyses of overall survival for all patients.

Variables	Univariate Analysis		Multivariate Analysis	
	HR (95% CI)	*p*-Value	HR (95% CI)	*p*-Value
Age, per year increase	1.02 (0.98–1.06)	0.319		
Comorbidity, no = 1	1.26 (0.6–2.65)	0.498		
Synchronous cancer, no = 1	2.91 (1.37–6.16)	0.005	2.19 (0.93–5.18)	0.074
Tumor length, per cm increase	1.01 (0.98–1.04)	0.633		
Tumor differentiation, poor = 1	1.18 (0.53–2.63)	0.686		
LVI and/or positive LN at esophagectomy, no = 1	0.997 (0.37–2.66)	0.995		
Pathological T-stage, pT1a-M3 = 1	3.61 (1.25–10.41)	0.017	3.09 (1.04–9.16)	0.042
R0 resection, yes = 1	2.55 (0.96–6.75)	0.06	3.51 (1.27–9.65	0.015
LN and/or distant recurrence, no = 1	3.45 (1.59–7.45)	0.002	2.13 (0.89–5.12)	0.09
Adjuvant therapy, no = 1	0.82 (0.25–2.74)	0.752		
Treatment method, ESD = 1	1.36 (0.64–2.91)	0.421		

Abbreviations: HR: hazard ratio; CI, confidence interval; ESD: endoscopic submucosal dissection.

## Data Availability

Deidentified individual participant data are available and will be provided on reasonable request to the corresponding author.

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
