# Peer review of "cT1N0M0 Esophageal Squamous Cell Carcinoma Invades the Muscularis Mucosa or Submucosa: Comparison of the Results of Endoscopic Submucosal Dissection and Esophagectomy"

_cancers, 2022, doi:10.3390/cancers14020424_

Round 1
Reviewer 1 Report
Dear authors!
Actually the manuscript is written quite well and the topic is very interesting for the clinical practice.
I have outlined a few concerns and questions:
1.) You stated, that the metachronous recurrance rate after ESD was significantly higher, than after esophagectomy. Twenty-one patients after ESD did not receive a CRTx. Is that probably the reason for the higher recurrance rate? Could it be worth to set up a study, where every patient after ESD receives a CRTx, independent of the depth of submucosa infiltration of the tumor (if an ethical committee vote is given)?!
2.) You did not show the Progression free survical rates. Can you show the data for the 2 groups?!
3.) The metachronous recurrance rate after ESD was 30% in your population. Despite that fact, the OS and DFS rates did not differ significantly between the two groups. Do you have an explanation for that?
4.) What influence does the R0 - resection rate of 87.5% after ESD have on the metachronous recurrance rate?
5.) What influence does the appearence of "synchronous cancer" have on the outcome after ESD or esophagectomy, condsidering the influence on the immune system?!
Author Response
Dear reviewer:
Thank you for your comments on our manuscript, because these valuable comments will greatly improve this manuscript.
Our replies are as the follows:
- In this study, we defined metachronous recurrence was a neoplasm that was detected at the esophagus other than the resection site (i.e., the scar area or surgical anastomosis) at least 6 months after the ESD/esophagectomy (line 148-150). While a neoplasm that was detected at the scar area/anastomosis of the esophagus was defined as local recurrence. This was because most of our patients with superficial esophageal cancer were caused by the habit of alcohol/smoking/betelnut chewing. Therefore, in chromoendoscopy, we often found multifocal Lugol-voiding lesions (LVLs) in most of our patients. We believed that LVLs (except for the main lesion resected by ESD) were responsible for the high rate of metachronous recurrence after ESD. Since we followed up endoscopy regularly for these patients, most metachronous recurrence were just high-grade dysplasia (75%). Therefore, we report metachronous recurrence and local recurrence separately. There is no evidence to support whether CRTx can reduce the risk of metachronous recurrence. Therefore, it might be interesting to conduct a study to prove it. However, the problem with radiotherapy is its radiation field (ie, the regional area adjacent to the post-ESD wound versus the entire esophagus). Instead, we have thought of using endoscopic radiofrequency ablation of all LVLs to reduce the risk of metachronous recurrence. But due to the high price (patients had to pay by themselves), we only have experience in a few cases. So we cannot make a solid conclusion.
- Our study subjects were cT1N0M0 ESCC patients. Therefore, after primary therapy (either ESD or esophagectomy), the patients would have no detectable disease. Therefore, when local/LN/distant metastasis occurred, the disease-free status and the progression-free status ended at the same time. In other words, the disease-free survival and the progression-free survival would be the same at this study. Therefore, we replaced disease-free survival (DFS)with progression-free survival (PFS) in the revised manuscript.
- As mentioned above, metachronous recurrence were high grade dysplasia (75%) and pT1a ESCC (25%), which could be cured by ESD. Therefore, we believed that it had no impact on the OS. Therefore, we did not take metachronous recurrence into the consideration of disease recurrence/progression. In other words, PFS was calculated based on local recurrence or LN/distant recurrence but not metachronous recurrence (line 161-163).
- As mentioned in the point 1, we believed that the R0 resection rate of ESD affects the local/LN/distant metastasis rate, but does not affect the metachronous recurrence rate.
- In our institution, we perform endoscopic screening of patients with head and neck cancers diagnosed recently or previously (that is, cured). Therefore, in this study, most of the patients who had synchronous cancer were actually head and neck cancers. Recently we have a study on synchronous head and neck cancers and superficial esophageal cancers. We found that the progression of head and neck cancers (usually, advanced stage (defined as stage III and IV)) was one of the two independent factors of OS for the patients with synchronous cancers (unpublished data).
Reviewer 2 Report
This paper is a retrospective look at comparing esophagectomy with ESD with very specific early esophageal cancers.
It's well written in text and grammar and has sufficient references. The graphs are understandable and appropriate
We suggest to restate your text from cT1b/pT1b to pT1b-SM1 in all areas. This really is the issue not T1b in general.
BTW in discussion you might discuss why only ESD patient with pT1b-SM1got CRT. Why didn't the esophagectomy patients receive any chemotherapy? According to your K-P curve they had no better disease free survival and higher lymph vascular invasion rate?
221: Please add a subgroup analysis of the patients who received ESD followed by CRT and compare their primary and secondary outcomes to the esophagectomy group.
Do you recommend all pT1b-SM1 get chemotherapy regardless of treatment type? Please discuss.
17: Define CRT abbreviation as this is the first time it is being used then can carry forward.
57: Elaborate more on the discrepancy of the diagnostic tools in differentiating between cT-stage and pT-stage in SESCC.
235: Correct RFS to DFS.​
Author Response
Dear Reviewer:
Thank you for your valuable comments on our manuscript, because these comments will greatly improve this manuscript.
Our replies are as the follows:
- As the definition of this study, clinical staging was mainly determined based on EUS results (line 90-97). EUS, In the best case, could only distinguish between cT1a or cT1b ESCC. It could not distinguish between T1b-SM1 and T1b-SM2. Therefore, regarding clinical staging, we could only describe cT1b in this manuscript, but not cT1b-SM1 or cT1b-SM2. For ESD specimens, the pathologists did report pT1b-SM1 or pT1b-SM2 according to the definition in this study (line 116-119). However, for surgical specimens, the pathologists reported only pT1b. We ever asked the pathologist to check the invasion depth of pT1b ESCC in surgical cases, but the results were not applicable because the submucosa depth was significantly wider than that in the ESD specimens (this may be due to the difference in preparation methods between esophagectomy and ESD). In the literature, the classification of submucosal tumor invasion depth of surgical specimens is to divide the submucosal layer by three. Therefore, in surgical case, we could not make sure whether the ESCC was pT1b-SM1 or pT1b-SM2 according to the definition (line 116-119). Therefore, in most cases, we could only mention pT1b in this manuscript.
- Our study subjects were cT1N0M0 esophageal cancer patients. As mentioned in the Introduction section, the main limitation of ESD is that it can only remove the primary tumor but not the metastatic lymph nodes (line 49-50). From surgical series, the risk of subclinical LN for cT1bN0M0 ESCC patients is 20–27%. Therefore, after ESD, some patients do require adjuvant therapy (either CRT or esophagectomy) to reduce the risk of LN recurrence (line 311-313). After ESD, we recommend additional therapy (esophagectomy or CRT) for the following patients: (1) positive vertical resection margins; (2) the presence of LVI in the resection specimens; and/or (3) any cancer with invasion ≥SM2. (line 122-126). However, LN dissection is part of an esophagectomy.
In our institution, we perform endoscopic screening of patients with head and neck cancers diagnosed recently or previously (that is, cured). Therefore, in this study, most of the patients who had synchronous cancer were actually head and neck cancers. Therefore, systemic chemotherapy aimed at treating patients’ synchronous cancer (such as head and neck cancer) was not regarded as an adjuvant therapy for esophageal cancer in this study (line 138-140).
- We analyzed the results between ESD combined with RT/CRT (n = 7) and esophagectomy (n = 32). The results are shown in Figure 2c. At the end of the follow-up, no significant differences were found in the OS rate (p = 0.999), DSS rate (p = 0.920) and PFS rate (p = 0.735) between the ESD combined with RT/CRT and esophagectomy groups. (line 248-252)
- According to our previous review article, chemotherapy alone is not recommended as an adjuvant therapy after the ESD (reference 30 in this manuscript). As mentioned in the point 2, only patients with risk of subclinical LN/distant metastasis are recommended to receive adjuvant therapy after ESD.
- We defined the CRT abbreviation when it first appeared in the revised manuscript. (line 58)
- Excluding 9 patients with uncertain clinical stage (ie, cT1 stage), the sensitivity, specificity, positive predictive value and negative predictive value of EUS for detecting pT1b ESCC were 31.7%, 77.3%, 72.2%, and 37.8%, respectively. (line188-191).
- Based on another reviewer’s suggestion, we delete the DFS and add progression-free survival (PFS) in the revised manuscript. (line 228-253)
Reviewer 3 Report
I want to thank for the opportunity to review the manuscript "cT1N0M0 esophageal squamous cell carcinoma invades the muscularis mucosa or submucosa: comparison of the results of endoscopic submucosal dissection and esophagectomy" submitted by Wang and co-workers.
I have read this paper with great interest because this manuscript addresses an important issue in thoracic surgery: how to deal with these patients with esophageal cancer cT1a and cT1b, respectively. Especially the latter one still represents a controversial subject between gastroenterologists and surgeons.
In my opinion the diagnostic and therapeutic procedure is well described, the manuscript is structured quite clearly and the statistical work-up seems to be sufficient from my point of view (but I´m a thoracic surgeon and not a qualified statistician).
However, I have some minor queries which have to adressed carefully:
1) I´m still wondering about the difference of LN and/or distant recurrence rate in both groups. Why is the LN-rate in the esophagectomy group higher than in the ESD group? What are the reasons for this astonishing result? The authors should clarify this aspect and provide more information about the intra-operative two field lymph node dissection.
2) Are there any data about the lymph node situation in the esophagectomy group before and after surgery? Was there any change from the pre-operative nodal staging (cN) to the definitive postoperative histological staging (pN)? I suppose that this might be those 4 patients mentioned from line 195 to 196?
3) I´m really surprised abot this enormous differences in R0 resection rates within the ESD collective. Is this acceptable?
4) In the limitations section the authors report about the lack of consensus on the selection criteria for adjuvant therapy after ESD. The authors should provide more information regarding the impact of an multidsiciplinary tumor board. I´m still missing this very important tool in this manuscript.
Author Response
Dear Reviewer:
Thank you for your valuable comments on our manuscript, because these comments will greatly improve this manuscript.
Our replies are as the follows:
- LN metastasis rate is related to the depth of tumor invasion. Therefore, the reasons for the higher LN and/or distant recurrence rate in the esophagectomy group might be: (1) There were significantly more pT1b ESCC patients in the esophagectomy group (87.5% vs. 52.5%, p = 0.001). The results might be caused by the selection bias we described in the discussion section (line 313-314). As shown in Table 1, cT1a patients tended to receive ESD rather than esophagectomy (77.5% vs. 43.8%). (2) An experienced EUS endoscopist (Mu-Hsien Lee) would review endosonographic images at the multidisciplinary meeting. For cT1b cases, if the EUS endoscopist judged the cases to be a superficial submucosal infiltration (irregular submucosal but not thinning or only mild thinning (compared to non-tumor parts)), these patients would be referred to ESD. (3) After ESD, patients at risk of LN metastasis or recurrence would receive adjuvant therapy (either CRT or esophagectomy, line 125-144). Of the 8 patients who underwent esophagectomy after primary ESD, 2 had pathological lymph node metastasis after surgery. These two patients were classified into esophagectomy group in this study. (4) More patients in the esophagectomy group had synchronous cancer. When the detectable LN/distant metastasis could not be confidently attributed to which primary cancer, we attributed the metastasis to ESCC. We added these points to the discussion section (line 296-307).
About the intra-operative two field (mediastinal and abdominal) lymph node dissection, the number of LN removed depended on the surgeon. The median of LN dissected for pathological examination was 25 (range, 2-49) (line 104-106).
- The subjects of our study were cT1N0M0 patients. Therefore, all patients were cN0 before treatment. Four patients (12.5%) in the esophagectomy group had pathological LN after the surgery, indicating the limitation of clinic staging tools even the use of PET-CT scans.
- ESD is a highly technique-demanding procedure. Therefore, it can be expected that the R0 resection will vary (greatly) among experienced and less-experienced endoscopists. Since we included 3 endoscopists having different degree of experience, the results of ESD in this study may be regarded as an averaged-experience endoscopist. However, we recommended esophageal ESD should be reserved for the experienced endoscopists in the discussion section (line 286-289).
- According to our institutional policy, only treatment-naïve patients must be discussed for their primary treatment in the multidisciplinary tumor board. Therefore, what kind of adjuvant treatment is required after ESD is determined by individual doctors, and usually depends on the patient's comorbidities or the patient's wishes (line 108-110, 122-126).